# Analysis of the Humoral Immunal Response Transcriptome of *Ectropis obliqua* Infected by *Beauveria bassiana*

**DOI:** 10.3390/insects13030225

**Published:** 2022-02-24

**Authors:** Yanhua Long, Tian Gao, Song Liu, Yong Zhang, Xiayu Li, Linlin Zhou, Qingqing Su, Letian Xu, Yunqiu Yang

**Affiliations:** 1School of Life Sciences, Anhui Agricultural University, Hefei 230036, China; longyanhua@ahau.edu.cn (Y.L.); zhangyong_zy888@163.com (Y.Z.); xyl@stu.ahau.edu.cn (X.L.); zll@stu.ahau.edu.cn (L.Z.); qqsu0121@163.com (Q.S.); 2State Key Laboratory of Tea Plant Biology and Utilization, Anhui Agricultural University, Hefei 230036, China; gaotian_gt5232@163.com (T.G.); liusong11@stu.ahau.edu.cn (S.L.); 3State Key Laboratory of Biocatalysis and Enzyme Engineering, School of Life Sciences, Hubei University, Wuhan 430062, China

**Keywords:** *Ectropis obliqua*, *Beauveria bassiana*, hemocytes, fat body tissues, immunity-related genes

## Abstract

**Simple Summary:**

*Ectropis obliqua* is a destructive leaf-eating pest that is widely distributed in China’s tea gardens. This pest shows remarkable resistance against multiple insecticides. As an environmentally friendly entomopathogen, *Beauveria bassiana* has been widely used to prevent agricultural pest infestations. However, the molecular mechanism of *B. bassiana* against *E. obliqua* remains unclear. We firstly isolated and identified a highly virulent *B. bassiana* strain. Using a transcriptome, we analyzed the differences of immune gene expression levels in fat bodies and hemocytes of *E. obliqua* that were infected by the *B. bassiana*, which provide molecular insights into the insect–pathogen interaction.

**Abstract:**

*Ectropis obliqua* is a destructive masticatory pest in China’s tea gardens. *Beauveria bassiana* as microbial insecticides can effectively control *E. obliqua* larvae; however, the immune response of this insect infected by *B. bassiana* are largely unknown. Here, after isolating a highly virulent strain of *B. bassiana* from *E. obliqua*, the changes in gene expression among different tissues, including hemocytes and fat bodies, of *E. obliqua* larvae infected by the entomopathogen were investigated using transcriptome sequencing. A total of 5877 co-expressed differentially expressed genes (DEGs) were identified in hemocytes and fat bodies, of which 5826 were up-regulated in hemocytes and 5784 were up-regulated in fat bodies. We identified 249 immunity-related genes, including pattern recognition receptors, immune effectors, signal modulators, and members of immune pathways. A quantitative real-time PCR analysis confirmed that several pattern recognition receptors were upregulated in hemocytes and fat bodies; however, others were downregulated. The investigated immune effectors (*ATT* and *PPO-1*) were suppressed. The results showed that there were tissue differences in the expression of immune genes. This study provides a large number of immunity-related gene sequences from *E. obliqua* after being infected by *B. bassiana*, furthering the understanding of the molecular mechanisms of *E. obliqua* defenses against *B. bassiana*.

## 1. Introduction

*Ectropis obliqua* (*Lepidoptera*: *Geometridae*) is the most harmful leaf-feeding pest of tea gardens in China, owing to its wide distribution and great destructiveness [1,2]. The larvae feed on tea leaves, consuming all the tea leaves in a tea garden during the outbreak period, badly inhibiting tea plant growth, and reducing tea production [3,4]. In recent years, insecticide-mediated chemical management methods have not only harmed human health and the environment, but they have also caused a more serious problem: insecticide resistance in pests [5,6,7]. Therefore, there is an urgent need to develop new pest control methods that are eco-friendly, sustainable, and able to overcome insecticide resistance [8].

*Beauveria bassiana*, an entomopathogenic fungus, has a wide host range, is environmentally friendly, and markedly improves pest control efficacies, resulting in its use as a biological control agent worldwide [9,10]. It has significant insecticidal effects on *Ectropis obliqua* [11], *Rhynchophorous ferrugineus* [12], *Spodoptera frugiperda* [13], *Tetranychus urticae* [14], and *Tuta absoluta* [15,16]. Generally, the infection mechanism of *B. bassiana* is as follows: the fungal conidia adhere to the host surface, proteases and chitinases are then secreted to hydrolyze the host cuticle, and then, the fungi invade and colonize the host hemocoel. They grow by absorbing host nutrients, and eventually the host is killed by the secreted toxins and physiological starvation [8,17,18,19,20]. The host’s natural cellular and humoral immune responses are activated after pathogenic fungi overcome the host’s physical barriers. In previous studies, a large number of immune genes have been identified in *Helicoverpa armigera* [21], *Clanis bilineata* [21], *Galleria mellonella* [22], *Riptortus pedestris* [23], *Ostrinia furnacalis* [24], and *Hypothenemus hampei* [25] infected with *B. bassiana*. These data provide clues and gene candidates for further exploration on the molecular mechanisms of entomopathogenicity by this fungus. However, the immune response mechanisms of *E. obliqua* after infection by the pathogenic fungi *B. bassiana* remain unclear.

In this study, we determined the sensitivity of fifth-instar *E. obliqua* larvae to the adherence of *B. bassiana* conidia, and then, the transcriptomic profiles of hemocytes and fat body tissues from *E. obliqua* infected by *B. bassiana* were analyzed using a high-throughput RNA sequencing method. We identified potential immunity-related genes through comparisons with homologous sequences of other insects. Finally, the expression levels of several key host immune genes were verified using quantitative real-time PCR (qRT-PCR). This study provides a theoretical basis for understanding the immune mechanisms of *E. obliqua* against *B. bassiana*, providing molecular insight into the complicated interaction between host and fungal pathogen.

## 2. Materials and Methods

### 2.1. Isolation, Cultivation, and Identification of Entomopathogenic Fungi

Fungal-infected *Ectropis obliqua* larvae was collected from tea plant leaves in the wild at Dayangzhen Tea Garden in Hefei, China (31.92° N, 117.21° E). Entomopathogenic fungi were isolated from the infected *Ectropis obliqua* larvae. Mycelium on the surfaces of *E. obliqua* were inoculated into potato dextrose agar medium and cultured in an incubator at 28 °C for 5 days. The isolated strains were purified three times. The benzyl chloride method was used to extract fungal DNA [26]. Using the genomic DNA of the strain as the template, the ITS-rDNA sequences were amplified using PCR with the universal primer pair ITS4 (5′-TCCTCCGCTTATTGATATGC-3′)/ITS5 (5′-GGAAGTAAAAGTCGTAACAAGG-3′). Briefly, the PCR reaction conditions were as follows: pre-denaturation at 95 °C for 5 min and 30 cycles of denaturation at 95 °C for 1 min, annealing at 55 °C for 1 min, and extension at 72 °C for 1 min, followed by a final elongation step at 72 °C for 10 min. PCR products were detected by electrophoresis in 1% agarose gels. Later, the target ITS sequences were submitted to National Center for Biotechnology Information (NCBI) and Basic Local Alignment Search Tool (BLAST) were used for database search, and the identified homologous sequences were downloaded. A phylogenetic tree was constructed using MEGA-11 software (Home (megasoftware.net, accessed on 6 December 2021)).

### 2.2. Insect Rearing and Infection Bioassays

*E. obliqua* moths were acquired from the State Key Laboratory of Tea Plant Biology, Anhui Agricultural University, Hefei, China (31.86° N, 117.27° E). The larvae were reared on tea leaves at 23 ± 2 °C and 70–80% relative humidity with a 16 h light/8 h dark photoperiod in the insect-rearing laboratory. Tea leaves used in the experiment were inserted into flower mud for storage. A total of 120 fifth-instar larvae were randomly selected and were injected with *B. bassiana* conidial suspension (2 μL, 1 × 10^7^ conidial mL^−1^) and Tween 80 solution (2 μL) using microliter syringes (Shanghai GaoGe Co., Shanghai, China) (*n* = 60). The numbers of dead larvae were recorded for 10 days. During this period, the dead larvae were collected and kept moisturized to observe whether the white hyphae could grow from the larvae.

### 2.3. RNA Preparation and Illumina Sequencing

At 0 h and 48 h of infection, fat body tissues and hemolymph samples of larvae were collected. Fat body tissues from ten larvae were collected under the microscope with dissecting forceps as one sample, respectively. Hemolymph collected from ten larvae was pooled into a 1.5 mL RNA-free microcentrifuge tube containing 0.1% 1-phenyl-2-thiourea and centrifuged at 500× *g* for 5 min at 4 °C to collect hemocytes [19]. Each sample was performed in triplicate. The samples were stored at −80 °C. Complementary DNA (cDNA) libraries were constructed using an Illumina Truseq™ Sample Prep Kit (Illumina, CA, USA). All the samples were sequenced on an Illumina Novaseq6000 platform (Illumina, San Diego, CA, USA). Original data were stored in FASTQ format.

### 2.4. Assembly and Annotation of Transcripts

Before assembly, to obtain satisfactory determination results, low-quality sequences (Quality < 20), sequences with nitrogen contents over 10%, and adapter sequences were removed. Trinity software was used to de novo assemble all the clean reads to produce contigs and singletons [27]. By comparing the known sequences in the public database, the functional annotation of these unigenes was obtained. The assembled unigene sequences were annotated against the NCBI non-redundant protein sequence and Swiss-prot databases using BLASTX (E-value < 10^−5^) [20,28]. The gene ontology (GO) annotation was obtained using the Blast2GO program [19,20]. The Kyoto Encyclopedia of Genes and Genomes (KEGG) annotation was performed to discover the potential enriched pathways [29].

Differentially expressed genes (DEGs) were calculated using the fragments per kilobases per million mapped reads (FPKM) method, and the screening conditions (|log2fold change| ≥ 1 and *p*-value < 0.05) were used to identify the DEGs [30]. Through a comparison with known amino acid sequences of immunity-related genes in other insects, the *E. obliqua* unigenes were obtained using the tblastn method. Subsequently, the *E. obliqua* immunity-related genes were manually confirmed.

### 2.5. Tissue Differential Expression Assessed by qRT-PCR

Another sixty *E. obliqua* larvae were randomly selected for a test of injection with *B. bassiana* conidial suspension (1 × 10^7^ conidial mL^−1^) and Tween 80 solution (*n* = 30). Ten larvae in each treatment were randomly chosen, and total RNA was extracted from fat body tissues and hemolymph. Each sample was performed in triplicate. In total, 1 μg of RNA was reverse transcribed using the StarScript II First-strand cDNA Synthesis Kit (GenStar, Beijing, China). Quantitative real-time PCR (qRT-PCR) was performed on an ABI 7300 Real-Time PCR System (Applied Biosystems, Foster City, CA, USA) using GoTaq qPCR Master Mix (Promega, Madison, WI, USA) in a 20-µL volume. Each well plate was loaded with 1μL cDNA. The reaction steps were as follows: 95 °C for 2 min and 40 cycles of 95 °C for 15 s and 60 °C for 30 s. The data were analyzed using the 2^−ΔΔCt^ method [20,31]. The *β-actin* gene of *E. obliqua* was used as the reference gene. The primers were described previously (Appendix A).

### 2.6. Data Analysis

GraphPad Prism 7.0, AI, Tbtools, IBM SPSS Statistics 25 was used for statistical analyses and to construct figures. A *t*-test was used to analyze the survival curve, and qPT-PCR data were analyzed using the 2^−ΔΔCt^ method.

## 3. Results

### 3.1. Biological Identification of B. bassiana and Its Virulence against E. obliqua

On the basis of the phylogenetic tree, the gene sequences of the target strain were clustered with those of multiple *Beauveria bassiana* strains (Figure 1 and Appendix A); consequently, the strain Bb-1 was identified as *B. bassiana*. Over 97% of *E. obliqua* fifth-instar larvae were killed by the *B. bassiana* infection after 60 h, which was significantly greater than the larvae killed by the Tween 80 treatment (*p* < 0.01) (Figure 2A). After the injection of *B. bassiana* spores, all the larvae died and became stiff after 60 h treatment, and the white hyphae on the surface of infected larvae were visualized at 4 days post-infection, which then covered the whole insect body after another 3 days of infection (Figure 2B).

### 3.2. Transcriptome Analysis after B. bassiana Infection

To obtain a transcriptome of *Ectropis obliqua*, the hemocytes and fat body tissues were collected from *B. bassiana*-infected larvae after 48 h. A total of 56,412 unigenes were annotated using BLASTX searches. A Venn diagram analysis indicated that 5877 genes were co-expressed between hemocyte and fat body samples (5826 upregulated and 51 downregulated in hemocytes and 5784 upregulated and 93 downregulated in fat bodies) (Figure 3, Appendix A). The DEGs in the hemocytes and fat bodies were enriched in the following GO terms: biological processes (cellular activity and metabolic process), molecular functions (catalytic activity and binding), and cellular components (cell part, membrane part, organelle, and protein-containing complex) (Figure 4).

To compare gene expression levels in the hemocytes and fat bodies of *B. bassiana*-infected larvae, a hierarchical clustering analysis of the DEGs was performed using fragments per kilobases per million mapped reads. Four major gene clusters were identified that exhibited distinct expression patterns among the different groups (Figure 5, Appendix A). The 482 genes in cluster 2 and the 24 genes in cluster 3 had higher and lower expression levels, respectively, in the hemocytes and fat bodies of *B. bassiana*-infected larvae. The 5275 genes in cluster 1 showed significantly greater RNA levels in hemocytes from entomopathogenic fungi-infected larvae compared with those of non-infected larvae. Cluster 4 (96 genes) showed identical expression patterns in the hemocytes and fat bodies of *B. bassiana*- and Tween-infected larvae. Four gene clusters were categorized into thirteen functional pathways using KEGG classifications. Most of the upregulated genes in clusters 1 and 2 were involved in immune responses, energy, and carbohydrate, lipid, and amino acid metabolic processes (Figure 6).

Furthermore, 249 *E. obliqua* immunity-related genes were identified from 56,412 unigenes, and they were compared with known protein sequences in *Drosophila melanogaster* [32], *Bombyx mori* [33], and *Helicoverpa armigera* [28]. The *E. obliqua* immunity-related genes were classified as recognition molecules and immune response effectors, as well as components of extracellular signal modulation and intracellular signal transduction on the basis of their functions. Intracellular signal transduction genes included members of the Toll, IMD, JNK, and JAK/STAT pathways (Figure 7, Appendix A). The differential expression analysis indicated that 16 genes (14 upregulated and 2 downregulated) were overexpressed in hemocytes and fat bodies (Figure 8).

In addition, we compared and analyzed some immunity-related genes of *E. obliqua* with those of other major insects, including members of Lepidoptera (*Bombyx mori* [33], *Plutella xylostella* [34,35], *Helicoverpa armigera* [36], *Lymantria dispar* [19], and *Hepialus xiaojinensis* [37]), Coleoptera (*Tribolium castaneum* [38] and *Dendroctonus valens* [20]), Hymenoptera (*Apis mellifera* [39]), and Diptera (*Anopheles gambiae* [40] and *Drosophila melanogaster* [40]). The immunity-related genes of *E. obliqua* (148) were similar to those of *L. dispar* (150). The number of immunity-related genes was highest in *D. melanogaster* (180) and lowest in *A. mellifera* (63) (Table 1)

### 3.3. Expression Patterns of Immune-Related Genes Verified by qRT-PCR

In total, ten immunity-related genes with differential expression patterns in the transcriptome were selected for qRT-PCR (Figure 9). After 48 h, the *B. bassiana*-infected larvae’s genes involved in the recognition of C-type lectin *CTL*-10 were markedly upregulated, whereas the peptidoglycan recognition protein *PGRP*-5 and *PGRP*-6 genes were dramatically downregulated in hemocytes and fat bodies. The *GNBP-2* gene was downregulated in hemocytes but upregulated in fat bodies. While *SPH-5* was upregulated, *serpin-7* was not differentially expressed in fat bodies. The immune response effector genes *ATT* and *PPO-1* were significantly downregulated in hemocytes and fat bodies. Among the Toll pathway genes, *SPZ 1–5* genes were all mostly up-regulated in hemocytes and down-regulated in fat bodies, according to our transcriptome data. *SPZ-2* gene was further verified by qRT-PCR, which was consistent with the transcriptome data (Appendix A).

## 4. Discussion

We analyzed changes in gene expression levels in different tissues, including hemocytes and fat body tissues, of *E. obliqua* infected with *B. bassiana* using a transcriptome. We identified 5877 DEGs in hemocytes and fat body tissues of *E. obliqua* larvae. This indicated that the pathogenic fungal invasion can profoundly provoke the immune system of *E. obliqua*. These results were consistent with the gene expression in *L. dispar* infected by *B. bassiana* [19]. The infection of *E. obliqua* larvae by *B. bassiana* activated a series of molecular responses; besides, *B. bassiana* may also consume carbohydrates and lipids, according to the GO and KEGG annotations. In total, we identified 249 immunity-related genes, which was more than identified previously in other insects. In total, 190, 233, 232, 149, and 244 immunity-related genes have been identified in *O. furnacalis* [41], *H. armigera* [28], *M. sexta* [42], *P. xylostella* [34], and *L. dispar* [19], respectively. These immunity-related genes are involved in recognition, signal modulation, and signal transduction, and some act as effectors. However, the Toll signaling pathway does not contain the plasmosin *pelle,* and the IMD signaling pathway does not contain the *TAK*-binding protein *TAB* in this insect. This result needs to be clarified in the genome data of *E. oblogua* in the future. Similarly, the Hop signal molecules that activate the JAK/STAT signaling pathway have not been identified in the *B. mori* genome [33]. Plasmosin *MyD88*, which activates NF-kB transcription factors in the Toll signaling pathway, has not been identified in the *P. xylostella* transcriptome [34].

Insects perceive the presence of pathogens through intracorporal pattern recognition receptors in combination with pathogen-associated molecular patterns, which activate immune responses. Insect pattern recognition receptors mainly include CTLs, β-1,3-glucan recognition proteins, and PGRPs. CTLs recognize pathogens, enhance host phenol oxidase activity levels, and activate encapsulation and melanization reactions [43,44]. The injection of gram-negative bacteria into *M. sexta* larvae induces the synthesis of CTLs and activates PO [45]. In this study, the *CTL-10* gene was upregulated in both hemocytes and fat body tissues, indicating that it plays important roles in the immune responses of *E. obliqua.* The β-1,3-glucan recognition proteins, known as GNBPs, recognize the glucan component of fungal cell walls and bind to gram-negative bacteria. For example, GNBPs of *D. melanogaster* are recognized by fungi to activate the Toll pathway [46]. The *GNBP-2* gene in this study was upregulated in fat body tissues and downregulated in hemocytes, and this pattern may result from the use of different signaling pathways in different tissues. PGRP-LC and PGRP-LE in *D. melanogaster* recognize the meso-diaminopimelic acid-type PGN in gram-negative cell walls, thereby activating the IMD signal transduction pathway [32]. As shown in the transcriptome data (Appendix A), all *PGRP1-9* were down-regulated in hemocytes and fat bodies after *B. bassiana* infection, and the expression of *PGRP-5* and *PGRP-6* were further confirmed by qPT-PCR results. Moreover, effector genes (such as *ATT* and *PPO*) were downregulated in all the samples infected by *B. bassiana*. We speculate that the 48 h sample time point allowed a sufficient amount of time for the release of entomopathogenic toxins.

After insects recognize pathogens, the protease activity signals of serine proteases (SPs) and their homologs (SP homologs, SPHs) are amplified step by step and finally transmitted to the Toll pathway’s extracellular ligands SPZ or PPO. The enzyme-linked reactions of SPs are usually regulated by SP inhibitors (serpins). These proteases play crucial roles in the reduction or amplification of immune signals [40]. In this experiment, *SPH-5*, *SPZ-2*, and *serpin-7* all showed upregulated trends in the immune responses of *E. obliqua*. Signal transduction molecules are the most common type of immunity-related genes, and those in the Toll and IMD signaling pathways mostly showed upregulated trends. In addition, in this transcriptome data, the *JNK* pathway is activated after pathogen recognition, leading to the activation of the transcription factor to turn on expression of antimicrobial peptides (AMP), such as cecropin and gloverin. Both gloverin and cecropin of antimicrobial peptides were down-regulated in the hemocytes after *B. bassiana* infection.

In this study, we chose only one time point (48 h post-infection) to profile and characterize the immune reaction of *E. obliqua* against *B. bassiana*, which is insufficient to fully understand when the insect immune is initially responding and how immune genes are temporally expressed to counter the invasion of *B. bassiana*. Furthermore, we aimed at identifying a highly virulent *B. bassiana* strain, profiling the immune system of *E. obliqua*, and partly revealing the immune reaction of the pest against the entomopathogen. The difference of the immune reaction in *E. obliqua* against high and low virulence strains is an interesting question and deserves investigation. In summary, our study provided new insights into the expression profiles of host defense genes in non-model insects exposed to pathogenic fungi, allowing us to further investigate the functions of antifungal immunity-related genes and to improve our understanding of host–pathogen interactions.

## 5. Conclusions

After the isolation of a highly virulent strain of *B. bassiana* from *E. obliqua*, we identified 249 immunity-related genes from *E. obliqua* infected by the *B. bassiana* using a transcriptome and compared their expression with those of homologous genes in other insects. Then, we confirmed the expression patterns of certain genes using qRT-PCR. Our study provides a molecular basis for the function studies of antifungal immunity-related genes and advances the understanding of host–pathogen interactions of *E. obliqu**a*.

## Figures and Tables

**Figure 1 insects-13-00225-f001:**
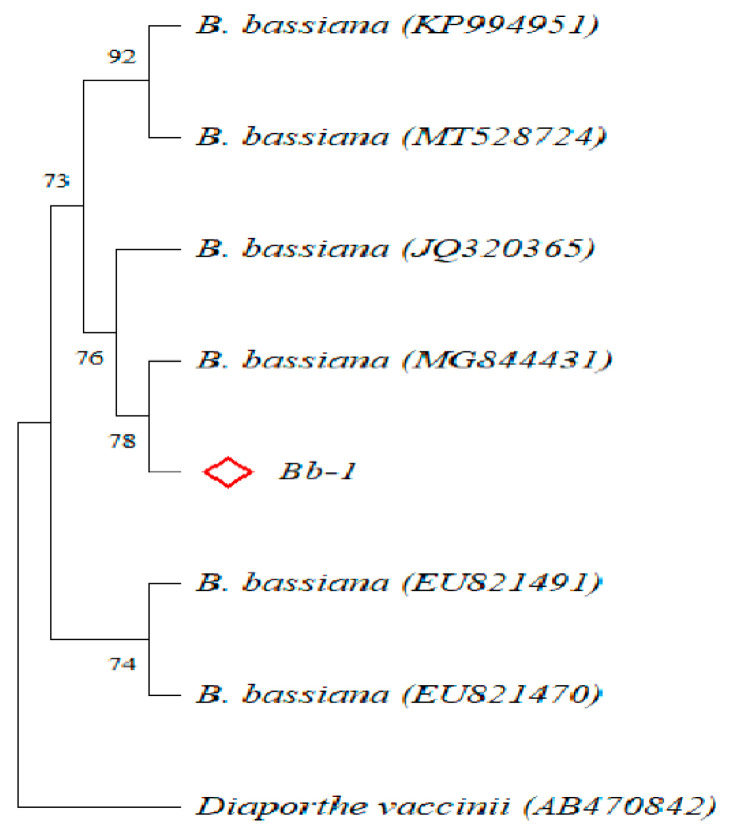
Construction of phylogenetic tree (Neighbour-Joining method; ◊ represent one *B. bassiana* strain isolated in this study; AB470842 *Diaporthe vaccinii* as outgroup).

**Figure 2 insects-13-00225-f002:**
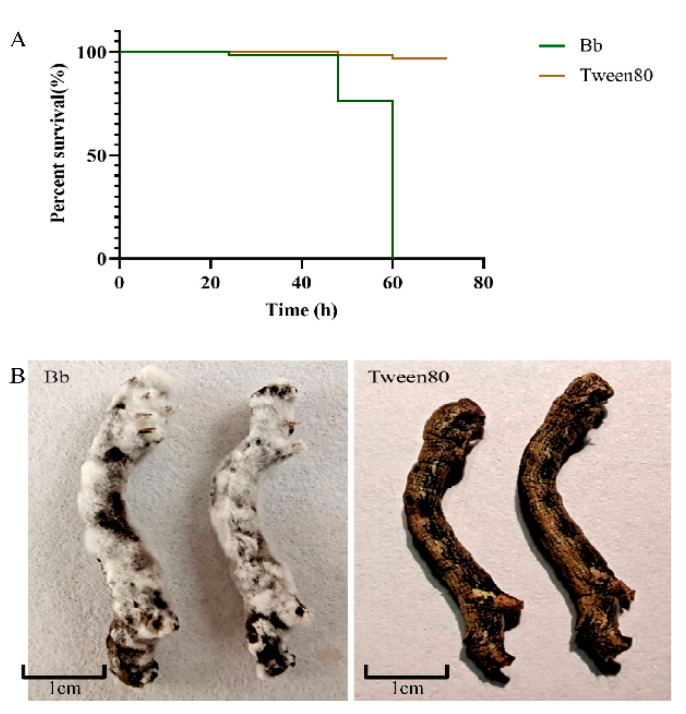
(**A**) Effects of injection on the survival rate of *E. obliqua* larvae. (**B**) Different phenotypes of the *E. oblique* larvae infected with *B. bassiana* and Tween 80.

**Figure 3 insects-13-00225-f003:**
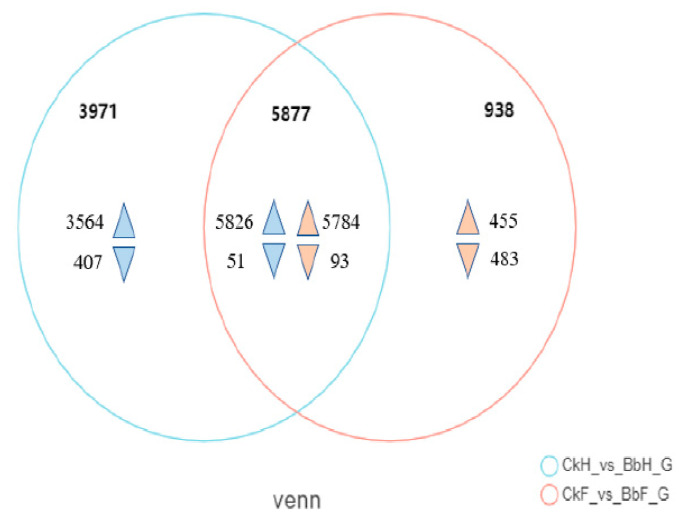
Venn diagram of the statistics of differentially expressed genes (DEGs) between fat body and hemocyte in *B. bassiana*- and Tween 80-treated *E. oblique* larvae.

**Figure 4 insects-13-00225-f004:**
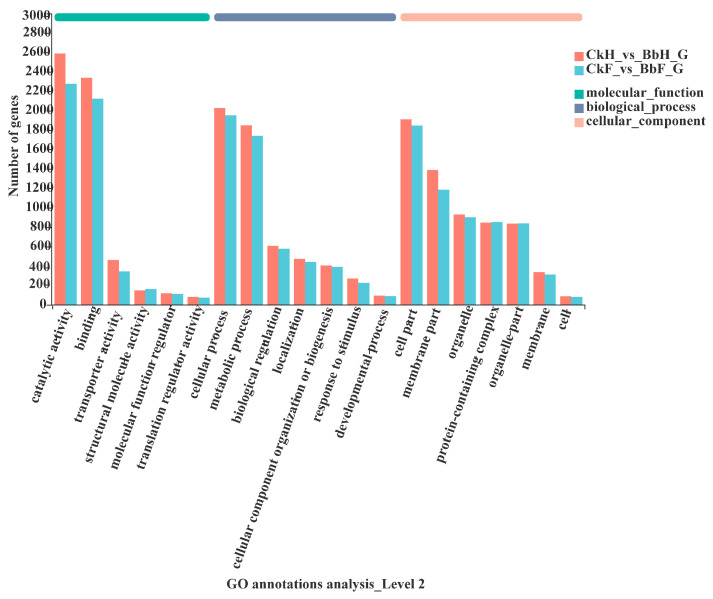
Gene ontology (GO) annotation of DEGs in the *E. oblique* transcriptome.

**Figure 5 insects-13-00225-f005:**
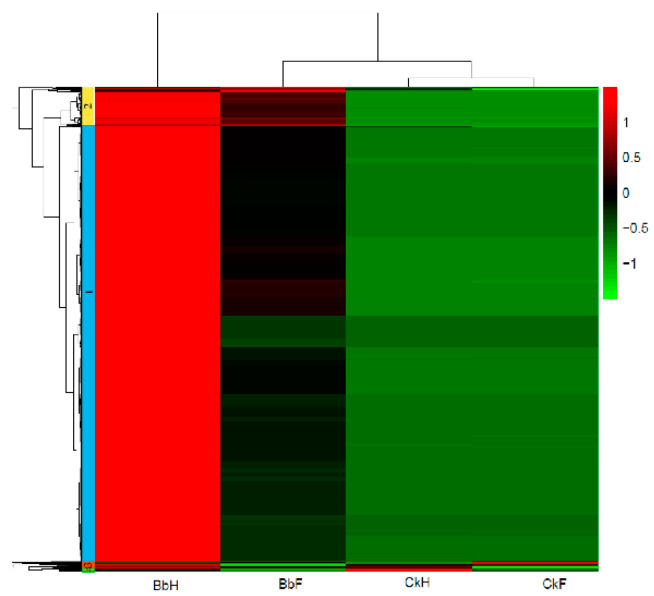
Hierarchical clustering analysis of mutual DEGs in fat body and hemocytes of *E. oblique* larvae.

**Figure 6 insects-13-00225-f006:**
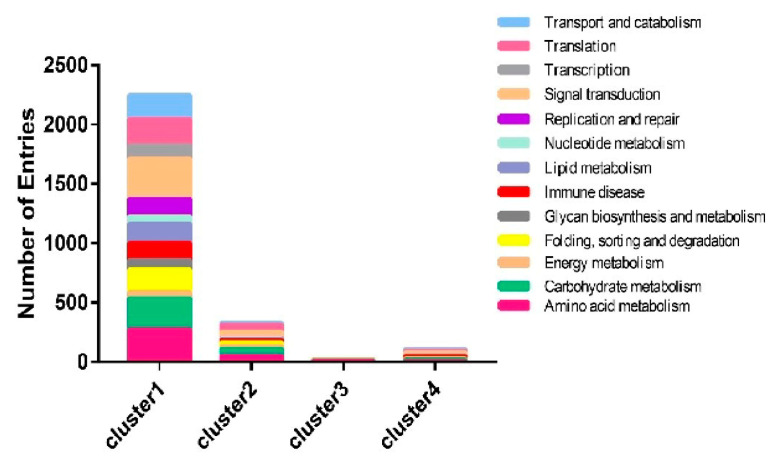
Functional classification of DEGs.

**Figure 7 insects-13-00225-f007:**
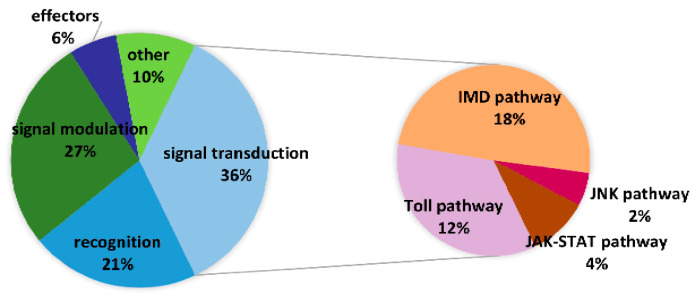
Distribution of *E. oblique* immunity-related genes in the categories of pathogen recognition, signal modulation, signal transduction (Toll, IMD, JNK, and JAK/STAT pathways), and immune.

**Figure 8 insects-13-00225-f008:**
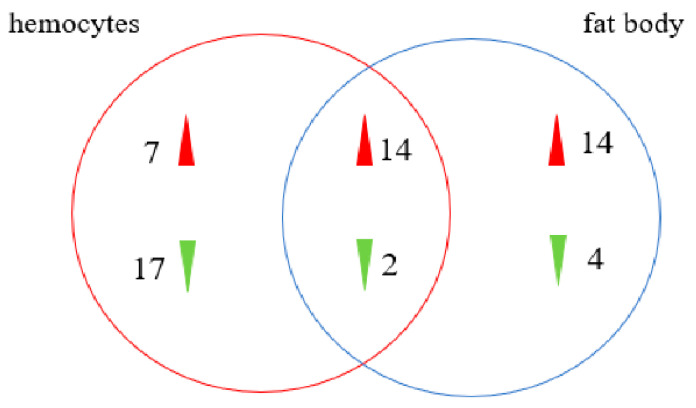
The immunity-related DETs in hemocytes and fat body after *B. bassiana* infection.

**Figure 9 insects-13-00225-f009:**
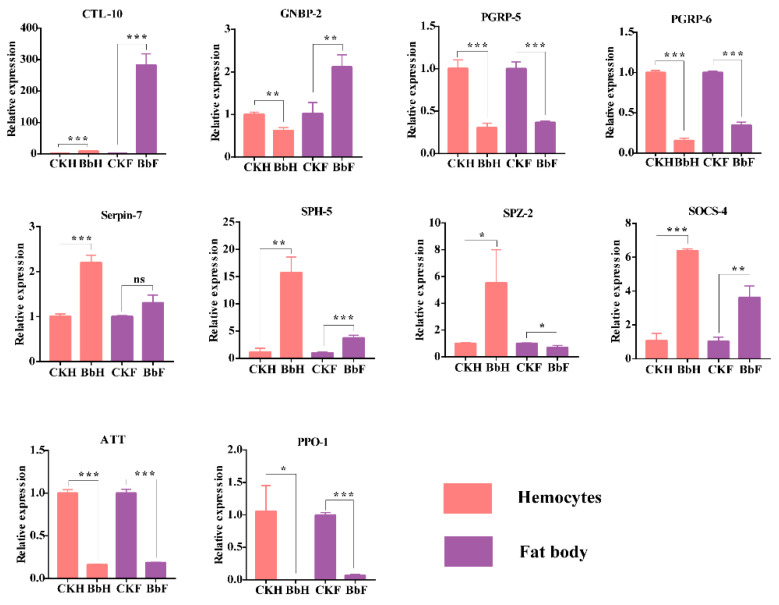
Quantitative real-time PCR analysis of the *E. oblique* immunity-related gene expression in hemocytes and fat body after *B. bassiana* (48 h) injection. *β-actin* was used as an internal reference gene. The data is represented as the mean ± S.D. (*n* = 3). * *p* < 0.05; ** *p* < 0.01. *** *p* < 0.001. ns: no significance.

**Table 1 insects-13-00225-t001:** Families and counts of innate immune genes from the insects. (The gene counts of *B. mori*, *T. castaneum, A. mellifera*, *A. gambiae*, *D. melanogaster* were based on genomic data.).

	*Lepidoptera*						*Coleoptera*		*Hymenoptera*	*Diptera*	
Gene Family	*E. obliqua*	*B. mori*	*P. xylostlla*	*H. armigera*	*L. dispar*	*H. xiaojinensis*	*T. castaneum*	*D. valens*	*A. mellifera*	*A. gambiae*	*D. melanogaster*
Recognition											
PGRP	9	12	9	9	10	9	7	6	4	7	13
βGRP/ GNBP	5	4	18	5	4	4	3	6	2	6	3
galectin	4	4	4	3	4	4	3	6	2	8	5
CTL	17	21	7	24	16	30	16	17	10	22	35
FREP	1	3	2	2	1	1	7	3	2	57	13
SR-B	11	13	13	10	9	7	16	6	10	16	12
TEP	3	3	1	3	4	3	4	6	3	15	6
modulation											
cSP/SPH	56	15	42	53	58	46	48	36	18	41	37
serpin	11	26	15	22	18	29	31	10	5	14	28
transduction											
cactus	1	1	1	1	1	1	1	3	3	1	1
Toll	16	14	9	11	12	5	9	6	5	11	9
SPZ	5	3	5	6	4	4	7	8	2	6	6
effectors											
PPO	2	2	1	2	3	2	6	3	5	5	8
other											
SOD	7	6	7	3	6	8	4	8	3	5	4
Total	148	127	134	154	150	153	162	124	74	214	180

## Data Availability

The data presented in this study are available in the Appendix A.

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
