# Peer review of "Analysis of the Humoral Immunal Response Transcriptome of Ectropis obliqua Infected by Beauveria bassiana"

_insects, 2022, doi:10.3390/insects13030225_

Round 1

Reviewer 1 Report

In this manuscript, Long et al. described some results regarding to the transcriptomic alterations of the hemocytes and fat body cells of a tea pest, Ectropis obliqua challenged by Beauveria bassiana infection, which may be related to the relationships between Entomogenous pathogens and host immunity. Many similar research articles have been published previously. The results from this manuscript were not like novel significance, just like a common sense. The interesting from this manuscript is very limited. At this version, the information from the results section and from the figures, especially, in the manuscript  are also very limited for the audiences. Additionally, both the English and scientific writings are not good. In my view, This manuscript does not meet the criteria for publication in the journal right now, so I will recommend rejection for it at this moment.  

Author Response

Thank you for your letter and  concerning our manuscript entitled “Analysis of the immune transcriptome of Ectropis obliqua infected by Beauveria bassiana (insects-1521351)”.

Reviewer 2 Report

  1. In line 127, the authors showed that obliqua fifth-instar larvae were killed by the B. bassiana after 60 h. How long the cascade larvae were covered with hyphae of B. bassiana, and which strain were you used in this study from the phylogenetic tree in Figure 1?
  2. A high virulence strain of bassiana was used in transcriptome analysis, and infected larvae were collected after 48h. So why did you analyze the data only at 48h? On the other hand, the low virulent strain of B. bassiana may be used to compare insect immune responses between high and low virulence strains. These might lead to identifying unique gene expression of insect immune and fungal virulence in high virulence strain-infected larvae. In addition, various infection times may be analyzed, such as 16, 24, 48, and 60 h, for understanding when the insect immune is initially responding and how positive or negative correlation with fungal virulence.
  3. In line202, Spätzle SPZ-2 gene in the Toll pathway was upregulated. How about other genes in the Toll pathway? In insect humoral immune, the Toll pathway is activated after pathogen recognition leading to activate transcription factor to turn on expression of antimicrobial peptides (AMP) such as gallerimycin and galiomycin in Galleria mellonella. Does it have any genes of AMP expression in obliqua?
  4. In line 245, “In this study, the PGRP-5 and PGRP-6 genes were downregulated in the immune responses, which may be why obliqua larvae were dead at 60 h after infection.” What does it mean? Why do PGRP-5 and PGRP-6 genes lead to larvae death at 60 h after infection?
  5. This study focused only on insect humoral immune responses. How about cellular immune responses such as hemocyte-specific genes?

Author Response

Dear reviewer:

Thank you for your letter and  concerning our manuscript entitled “Analysis of the immune transcriptome of Ectropis obliqua infected by Beauveria bassiana (insects-1521351)”. Those comments are all valuable and very helpful for revising and improving our paper, as well as the important guiding significance to our researches. We have studied comments carefully and have made correction which we hope meet with approval.

Reviewer 3 Report

This study is interesting but lacks too much of technical information to be properly reviewed. I hope my comments will help the authors improve their manuscript so the review process can be efficient.

The Introduction is concise and talks only about the need for biological control, but does not mention previous studies having analyzed transcriptomes or proteomes of insects infected with entomopathogenic fungi. I believe such references are needed in the intro.

Materials and methods : This section really lacks of information.

2.1. the reader needs to know how the authors isolated the strain in detail. Was the model insect exposed to some soil ? To some plant ? Was it used as bait or just wild collected ? This is not enough detail.

2.2. how were the insects sprayed ? On some filter paper ? On leaves ? With what device ? How long did the authors follow survival for ? What did they do with the cadavers ?

2.3. “anatomized” does not mean anything. How were the fat body and hemolymph collected ? With what device ? How were hemocytes purified from the hemolymph ?

If I understand correctly, there are only 3 biological replicates consisting in pools of 10 larvae. Why this choice ?

Unfortunately I am not able to judge the 2.4 section as my knowledge int his regard is limited.

2.5.: where was the RNA taken from ? The same insects which went through RNAseq ? Was it leftover RNA or were new insects extracted ?

What quantity of cDNA was loaded in each well of the plate ?

There is no Statistics section. The delta-delta-Ct method is not a statistical analysis per se.

In the Results section:

3.1.: How long did it take for the larvae to be covered in white mold ? What proportion of them was covered in white mold ?

I do not understand the caption of the figure 2A: was there an injection performed ? I thought the larvae had been sprayed.

Implementing these technical details in the manuscript would improve the reviewing process.

Author Response

(The authors gave the same response as above.)

Round 2

Reviewer 1 Report

I still stick my point of the view in the first round reviewing. I recommend rejection for this manuscript.

Author Response

Response to reviewer 1

Dear reviewer:

Thank you for your letter and  concerning our manuscript entitled “Analysis of the immune transcriptome of Ectropis obliqua infected by Beauveria bassiana (insects-1521351)”. We considered them carefully.

Point 1: Many similar research articles have been published previously. The results from this manuscript were not like novel significance, just like a common sense. The interesting from this manuscript is very limited.

Resopnsse: Thanks for your comments, which is highly appreciated. As for the novelty, it is well-known that many similar research articles about the relationships between entomogenous pathogens and host immunity have been published previously. However, the use of entomogenous fungi to control Ectropis obliqua, which is a destructive masticatory pest in China’s tea gardens, is rarely reported. This study provides a molecular basis for the use of entomogenous fungi to control E. obliqua, although such general studies go a long way in providing a molecular basis. Thanks again to the reviewer on suggesting to further improve this manuscript,we have studied comments carefully and have made corresponding corrections which we hope meet with approval.

Point 2 : At this version, the information from the results section and from the figures, especially, in the manuscript are also very limited for the audiences. Additionally, both the English and scientific writings are not good.

Resopnsse: Thanks for your kind suggestion. We have revised the results section and part of the Figure to allow the audiences to better understand our research. In addition, we have made extensive revisions to English and scientific writings. We appreciated for your warm work earnestly,and hope that the correction will meet with approval. Once again, thank you very much for your comments and suggestions.
